# *Helicobacter pylori* Prevalence and Risk Factors in Three Rural Indigenous Communities of Northern Arizona

**DOI:** 10.3390/ijerph19020797

**Published:** 2022-01-12

**Authors:** Robin B. Harris, Heidi E. Brown, Rachelle L. Begay, Priscilla R. Sanderson, Carmenlita Chief, Fernando P. Monroy, Eyal Oren

**Affiliations:** 1Department of Epidemiology and Biostatistics, Mel and Enid Zuckerman College of Public Health, University of Arizona, 1295 N. Martin Ave., Tucson, AZ 85724, USA; heidibrown@arizona.edu (H.E.B.); rlbegay@arizona.edu (R.L.B.); 2Department of Health Sciences, College of Health and Human Services, Northern Arizona University, SAS (Bldg 60), 1100 S. Beaver St., POB 15095, Flagstaff, AZ 86011, USA; priscilla.sanderson@nau.edu; 3Center for Health Equity Research, College of Health and Human Services, Northern Arizona University, 1395 South Knoles Drive, POB 4065, Flagstaff, AZ 86011, USA; carmenlita.chief@nau.edu; 4Department of Biological Sciences, College of the Environment, Forestry and Natural Sciences, Northern Arizona University, 617 S. Beaver St., Flagstaff, AZ 86011, USA; fernando.monroy@nau.edu; 5Division of Epidemiology & Biostatistics, School of Public Health, San Diego State University, 5500 Campanile Drive, San Diego, CA 92182, USA; eoren@sdsu.edu

**Keywords:** *Helicobacter pylori*, health disparities, American Indian, gastric cancer

## Abstract

*Helicobacter pylori* (*H. pylori*) is one of the most common bacterial stomach infections and is implicated in a majority of non-cardia gastric cancer. While gastric cancer has decreased in the United States (US), the incidence in the Navajo Nation is nearly four times higher than surrounding Non-Hispanic White populations. Little is known about *H. pylori* prevalence in this population or other Indigenous communities in the lower 48 states. In this cross-sectional study, 101 adults representing 73 households from three Navajo Nation chapter communities completed surveys and a urea breath test for active *H. pylori*. Accounting for intrahousehold correlation, *H. pylori* prevalence was 56.4% (95% CI, 45.4–66.8) and 72% of households had at least one infected person. The odds of having an active infection in households using unregulated water were 8.85 (95% CI, 1.50–53.38) that of the use of regulated water, and males had 3.26 (95% CI, 1.05–10.07) higher odds than female. The prevalence of *H. pylori* in Navajo is similar to that seen in Alaska Natives. Further investigation into factors associated with prevention of infection is needed as well as understanding barriers to screening and treatment.

## 1. Introduction

*Helicobacter pylori* (*H. pylori*) is among the most common bacterial stomach infections, with nearly half of the global population estimated to be infected [1,2,3]. Prevalence estimates vary considerably by age, race/ethnicity, and socioeconomic status (SES) [4,5,6]. Higher prevalence rates in older age groups may represent a cohort effect resulting from poorer living conditions during childhood [7]. A negative association is observed regularly between SES and *H. pylori* infection worldwide and in the United States (US) [1,3,8]. However, while *H. pylori* infection is asymptomatic in most individuals [9], especially early in infection [10], the infection has been implicated in the etiology of several severe gastrointestinal outcomes, including peptic ulcers, chronic gastritis, and gastric cancer [1,2,3]. While a minority of chronically infected individuals develop gastric cancer [11], *H. pylori* infection has been attributed to nearly 89% of all non-cardia gastric cancers, which it represents over three-quarters of all gastric cancer [12]. The recent World Cancer Report noted that infection with *H. pylori* is a necessary but not sufficient cause for gastric cancer [13].

Gastric cancer incidence and mortality are declining globally and, in the US [13,14,15]. However, while rates have decreased among American Indian/Alaska Native (AI/AN) populations (40% lower and 29% lower incidence in women and men, respectively between 1999 and 2015), the burden of the disease remains high [16]. For Alaska and the southwestern US, the relative risks for stomach cancer are much higher among AI/AN populations compared to White [16]. A county-level ecological study showed non-cardia gastric cancer mortality rates were at least two standard deviations above national average for counties in Alaska, New Mexico, and northern Arizona [17]. The Navajo Epidemiology Center reported the age-adjusted incidence rate of gastric cancer among AI/AN residents in the six US counties that overlap the Navajo Nation to be 14.2 per 100,000, which is nearly four times higher than the non-Hispanic white population in Arizona [18].

Risk factors for gastric cancer are multi-factorial and include the genetic profile of the host, characteristics of the *H. pylori* strains, and environmental factors that can include diet, water quality, and socioeconomic factors [13]. Possible reasons for the decline in gastric cancer rates include decreased intake of smoked, pickled, or salt-preserved foods, increased fruit and vegetable consumption, greater use of refrigeration, improved water sources, along with a decline in *H. pylori* prevalence [19,20]. Reasons for the high rates of gastric cancer in AI/AN are likely to include the risk factors named above as well as tobacco use and environmental factors such as source of the water supply [19,20,21,22]. While uncertainty remains regarding mode of transmission, *H. pylori* is believed to spread from person-to-person through a fecal-oral route [23]. Drinking water testing positive for *H. pylori* is associated with clinical symptoms [24] and an increased rate of gut colonization [25]. Living conditions in the US and the Navajo Nation, while significantly improved over past decades, still include crowded and substandard housing and may lack in-home piped water and waste removal.

Despite the elevated burden of gastric cancer in certain communities and populations and the established link between *H. pylori* infection and non-cardia gastric cancer, data on *H. pylori* prevalence in US AI/AN communities are sparse. Most data come from work in Alaska and Canada, where infection rates range from 38% to 95% [26,27], although a survey in a Montana Native American community reported a 53% prevalence [28]. Serum immunoglobulin G assays estimated prevalence to be around 75% in Alaska Natives [29] compared with 35% for the broader US [30]. A study of patient data of the prevalence of *H. pylori* infection in people experiencing upper gastrointestinal symptoms also found the highest prevalence of *H. pylori* infection in AI/AN [31]. These findings suggest a need for better understanding of what are likely high infection rates in rural and underserved communities in the US.

Beginning in 2016, investigators from two Arizona universities joined efforts as part of the Partnership for Native American Cancer Prevention (NACP) partnership to study the potential impact of *H. pylori* among the Navajo Nation. This partnership led to the Navajo Healthy Stomach Project, with the goal to assess the feasibility of conducting random household recruitment to estimate the prevalence of active *H. pylori* infection among Navajo adults and to identify social, behavioral, and environmental factors associated with infection. This information is needed in order to develop strategies to reduce the incidence of gastric cancer in high-risk populations.

## 2. Materials and Methods

### 2.1. Study Design and Participants

A cross-sectional study was conducted from June to August 2018 among Navajo citizens living in three chapters (local governmental regions of the Navajo Nation) or communities of northeastern Arizona. Participants were recruited from randomly selected households from each of the three chapters, with the goal to recruit 70–75 households representative of the selected chapters. These three chapters comprised approximately 978 square miles, with an average population density of 2.5 to 6.6 per square miles and 1278 occupied housing units according to the 2016 American Community Survey [32]. Household inclusion criteria were residence within chapter boundary areas, at least one household resident 18 years of age or older who self-identified as Navajo, and physically and mentally able to complete survey assessments and a urea breath test (UBT) to test for active *H. pylori* infection. Once one member of the household was recruited, other household members 18 years or older were eligible to participate and complete the survey and UBT.

### 2.2. Institutional Approvals

Prior to beginning any recruitment, formal community approvals were obtained via support resolutions from all three participating chapters and the two Navajo Agency Council areas that included their jurisdictional chapters. Securing these resolutions was a requirement prior to going to Navajo Nation Human Research Review Board (NNHRRB) for protocol approval. The University of Arizona Institutional Review Board also approved the final protocol, questionnaires, and consent forms. The NNHRRB also reviewed and approved the manuscript prior to submission for publication.

### 2.3. Selection of Households

Household recruitment goals for each chapter were established to assure that the overall recruitment would reflect the underlying population of the three chapters. First, using Google Earth, all household-like structures within the boundaries of the three chapters were visually pinned and locations exported to an Excel spreadsheet. Households were selected using a random number generator in Microsoft Excel and reverse geocoded for directions to the location. Of the 1384 potential household structures identified, 166 structures were randomly selected for recruitment. The 166 structures then were ground-truthed (a team member drove to the locations identified on Google Earth) which identified 143 inhabited houses. These houses became the sampling population, and all were visited for eligibility and interest. If individuals were not home, information about the study was left at the house (doorknob recruitment material and a flyer describing the study). If individuals were home, the study was explained to them and availability of interview times and names for re-contact were collected. Potential participating households were grouped by proximity to maximize coverage and driving time efficiency during the interview process.

### 2.4. Recruitment

Potentially eligible households were approached at least five times, at different times of day and days of the week, to ascertain if the household was eligible and/or interested in participating. Recruitment materials included description of the study, frequently asked questions, and a copy of the consent form. All recruitment materials had been reviewed for cultural appropriateness by Navajo investigators on the research team, Navajo Community Health Representatives (CHRs), and the Navajo Nation Human Research Review Board. A video was also created to show the UBT sample collection process and was available for viewing if participants requested additional information about the process. If the primary household contact and household members expressed interest in the study, the consent form was read to them and if still willing to participate, the consent form was signed and a copy provided for their records. A tracking form was used to assess workload and to calculate response rate. A $25 cash incentive for individual participation was offered to each participant at the end of the visit as well as the return of *H. pylori* test results.

### 2.5. Training

All research team members completed Human Subjects certification trainings prior to commencement of fieldwork. A 6-h household data collection training session was held on the first field day. Field personnel included the Navajo Healthy Stomach Project research team members, undergraduate student researchers from Northern Arizona University, the University of Arizona and Diné College, and Navajo CHRs. Personnel were assigned to a field team of 2–3 individuals, which included at least one CHR and one University staff or student. At least one member of the team was fluent in Navajo (usually the CHR) to translate instructions or administer the surveys as necessary. Each field team received a Training Protocol binder that included information about the pathogen, study aims, goals, copies of all data collection instruments or surveys, and instructions for consenting and interviewing participants, administering the UBT, and collecting water samples. Interviewers were trained on household interviewing techniques by the study team and on administration of the UBT by a nurse at the Winslow Indian Health Care Center (WIHCC). The total time for administering questionnaires and collecting the breath and water samples was between 1.5 and 2 h.

### 2.6. Data Collection

Field teams explained all study procedures, obtained written consent, administered all surveys, and collected specimens. The household survey was completed by one household member and assessed household characteristics including: number of people living in the household, water sources, food preparation and refrigeration practices, and latrine type. All consenting participants within the household also completed an individual survey which assessed: demographics, socioeconomic factors, lifestyle behaviors, dietary habits and behaviors, general health history, stomach symptoms, use of medication, and knowledge and perceptions of *H. pylori* and stomach cancer.

### 2.7. Measures

Active *H. pylori* infection was determined using a non-invasive C-Urea Breath Test (BreathTek UBT, Otsuka America Pharmaceutical, Inc., Rockville, MD, USA). UBT assesses the amount of labeled carbon dioxide exhaled after consuming a mixture including 13C-labeled urea, which is broken down in the stomach into carbon dioxide and other products by *H. pylori*. The test is fast, with high diagnostic accuracy (sensitivity = 96%, specificity = 93%) [33]. Exclusion criteria for testing included pregnancy and current or recent use of antibiotics and/or proton pump inhibitors. Breath specimens obtained from each participant were processed at the WIHCC gastroenterology clinic. The results from the UBT were reported as positive or negative for active *H. pylori* infection. If the breath specimen was too small to produce a conclusive result or an error occurred, the results came back indeterminant and were not included in the analysis.

UBT results were provided to participants along with general study findings and information on preventing *H. pylori* infection. Participants who tested positive received a recommendation to see their own health care provider, as well as the name and contact information for the WIHCC gastroenterology team. Navajo CHRs visited all participants to confirm each individual received a result and provide any further education.

The primary covariates of interest were self-reported individual and household level demographic, environmental, and clinical factors identified in the literature as commonly associated with *H. pylori* infection among AI/AN populations. Individual demographic characteristics were age (<50, ≥50), sex (male or female), and education (<high school, ≥high school diploma/GED). Clinical factors were previous history of ulcers (yes, no) and previous history of gallstones (yes, no). Household factors included household water source (regulated, unregulated, mixed), availability of electricity, refrigerator, and distance to grocery store. Regulated water was defined as municipal water direct piped to the home and/or bottled water delivered to the home or from the store. Unregulated water included water from a natural spring, community spigot, windmill, and/or private well. Household water that was a combination of these two sources was defined as mixed. *H. pylori* awareness questions were also asked and included having heard of the pathogen prior to this study (yes, no), having a family member testing positive (yes, no), themselves being tested (yes, no), and any follow up gastroscopy/endoscopy (yes, no).

### 2.8. Statistical Analysis

Individual and household characteristics were summarized by participant UBT result (positive or negative) and as overall totals. Prevalence of *H. pylori* infection was estimated for individual and household levels by calculating the proportion of positive UBT results among all definitive UBT test results. Comparisons between UBT result groups were made using chi-square tests. Logistic regression was used to calculate odds ratios (OR) and 95% confidence intervals (95% CI) as the measure of effect. To account for the intrahousehold correlation due to multiple participants within the same household, mixed effects logistic regression with household as the random effect was calculated. All statistical analyses were performed using R version 3.5.1. Model covariates were included based on prior published literature.

### 2.9. Sensitivity Analyses

Subset analyses were evaluated to look at potential impact for re-categorization of participant age and sex-specific factors. Age was re-categorized into five categories: 18–29, 30–49, 50–64, 65–79, and >80. Multiple and mixed effects logistic regression models as previously described were constructed.

## 3. Results

Figure 1 shows 73 households were recruited, for a 56% response rate (48%, 58%, 64%, in the three chapters). From these 73 households with a primary household contact participating, 106 individuals consented to completing an individual questionnaire and the urea breath test for active *H. pylori* infection. Only individuals that completed all assessments and had definitive *H. pylori* test results (*n* = 101) were included in these analyses. Of the 73 households, 25 (34.2%) had more than one person participate.

### 3.1. Individual and Household Characteristics

Most households in this study had electricity and refrigeration (*n* = 66 of 73, 90.1% for both). While most were connected to regulated (piped) or used bottled water (*n* = 53, 74.7%), 19.2% (*n* = 14 households) reported use of unregulated spring or well water and 6.9% (*n* = 5) reported mixed water sources. Households were relatively remote from resources, as indicated by 83.6% reporting travel times of more than one hour for groceries (*n* = 61 households).

The characteristics of the 101 participants with definitive *H. pylori* infection results and completing both the household and individual surveys are shown in Table 1. Participants were primarily female (*n* = 60, 59.4%), between 50 and 79 years of age (*n* = 54, 53.5%), and had at least a high school diploma or GED (*n* = 64, 63.4%). Previous history of ulcers was uncommon (*n* = 4, 4.0%) while nearly a quarter (*n* = 25, 23.8%) of participants had a previous history of gallstones. In addition, over 24% of participants reported a prior gastroscopy/endoscopy, 18.2% of those with UBT positive test and 34.3% of those with a negative test (*p* < 0.08). There were no observed differences between the three communities on these characteristics, thus results are provided in aggregate.

### 3.2. Prevalence of H. pylori Infection

Active *H. pylori* infection was found in 66 of the participants, with an overall crude prevalence of 65.3% (95% CI, 52.5–72.8). The prevalence was higher for males 78.0% (95% CI, 62.3–89.4) compared to 56.7% (95% CI, 43.21–69.4) in females and these differences were statistically significant (*p* = 0.03). The overall adjusted prevalence of infection was 65.4% (95% CI, 53.3–77.4) after accounting for correlation at the household level. Of note, over 72% of households had at least one person with a positive UBT. Of the 25 households with multiple participants, 23 households had two participants, one household had three participants, and one had four participants. Fifteen of the two participant households were UBT-result concordant, with individuals in 13 of these 15 households both testing positive.

### 3.3. Association of Risk Factors for Infection

Table 2 shows crude and adjusted associations between UBT status and potential individual and household level risk factors. Model 1 (AOR_1_) controlled for other risk factors in the table and Model 2 (AOR_2_) adjusted for those same factors and for multiple participants in the household. Participants whose household water source was unregulated had higher odds of an active *H. pylori* infection than participants whose household water source was regulated in both unadjusted and adjusted models (AOR_1_ = 8.61, 95% CI, 1.45–51.05). After accounting for intrahousehold correlation, the odds of having an active *H. pylori* infection in households with unregulated water compared with regulated water increased (AOR_1_ = 8.85, 95% CI, 1.50–53.38) as did the odds of infection among males compared with females (AOR_2_ = 3.26, 95% CI, 1.05–10.07). All other potential risk factors were not statistically significant in any of the models.

The adjusted models with re-categorized age showed similar results to the dichotomous analyses. After accounting for intrahousehold correlation, unregulated water (AOR = 9.41, 95% CI, 1.36–42.54) remained significantly associated with a positive UBT, as did being male (AOR = 3.21, 95% CI, 1.04–9.95). For the sex-specific analyses, the magnitude of the ORs were similar; however, the confidence intervals were extremely wide.

### 3.4. H. pylori Awareness

Few of the participants in this study had heard of (*n* = 24, 23.8%) or been tested for (*n* = 13, 12.9%) *H. pylori*. Table 3 shows the relationship between awareness of the *H. pylori* and UBT status. There was a non-significant trend where individuals testing negative by the UBT had heard of *H. pylori* and had an endoscopy. However, prior treatment history was significantly more common among those with negative results compared with positive UBT results (*p* < 0.01). There were too few individuals for stratified analyses by UBT result.

## 4. Discussion

*H. pylori* is one of the most common bacterial stomach infections worldwide with variations in prevalence occurring geographically and by levels of socioeconomic development [1,2,3,4,5,6]. High prevalence of infection is associated with housing and living conditions, age, gender, and socioeconomic status. Indigenous communities in South America, Canada, and Alaska have all been identified as having higher burden of *H. pylori* infection compared to non-Indigenous communities in the same geographic region [4,34,35,36,37]. Despite this knowledge, surrounding prevalence of infection and risk for gastric sequelae, factors for infection among Indigenous communities in the US are not well known. Herein, we determined *H. pylori* infection prevalence among Navajo communities in northern Arizona and identified household and individual level factors associated with infection.

This study in a rural section of Northern Arizona showed high prevalence of active *H. pylori* infection, 63.2% (95% CI, 52.5–72.8) before adjustment and 56.4% (95% CI, 45.4–66.8) after accounting for intrahousehold correlation. Further, over 72% of the households interviewed had at least one person who was positive for the infection. Most prior data on *H. pylori* prevalence in Indigenous communities of North America come from Alaska and Canada, where infection rates range from 38% in biopsy samples from northwestern Ontario [26] to 95% among 306 serum samples from a population in northwestern Manitoba [27]. Perhaps most consistent with the Navajo Healthy Stomach Project study design and population, a large study from among primarily rural living Alaskans (*n* = 710) using both UBT and serum samples, found high concordance between the two tests and estimated *H. pylori* prevalence to be 69% and 68%, respectively [34].

The Navajo Healthy Stomach Project identified water source as a primary environmental or household risk factor for infection. While uncertainty remains regarding mode of transmission, analysis of a national US database indicated *H. pylori* may be spread person-to-person through a fecal-oral route [23], potentially via contaminated water sources. *H. pylori* has been shown to survive in drinking water [22] and sources positive for *H. pylori* have been associated with clinical symptoms [24] and an increased rate of colonization [25]. While 19.2% of the households reported using only unregulated water sources (meaning hauling water from other locations), this percentage is down from 30% reported in an Environmental Protection Agency (EPA) report in 2003 [38]. Because of infection persistence, the infections observed in this study might have been obtained in previous years when in-home piped water was not available.

In the adjusted model, we estimated that men had 3.26 the odds of a positive UBT test compared with females. This sex association is stronger than that found from other studies, including two meta-analyses of adult male *H. pylori* infection prevalence (OR = 1.12, 95% CI, 1.09–1.15 and OR = 1.33, 95% CI, 1.04–1.70) [39,40]. This higher infection prevalence among men, however, is also observed in gastric cancer rates, where men have higher incidence and higher mortality rates than women across all race/ethnicity groups [41,42].

In contrast to other findings, this project showed that lower education and increased age were not associated with increased prevalence and odds of active infection [1,3,4,5]. However, due to the high infection prevalence in this sample it is difficult to determine if this is a true association or if the population in general had higher levels of education than anticipated or if environmental factors, e.g., the need to haul water for the family, or multigenerational family structures, are stronger risk factors.

The Navajo Healthy Stomach Project also found that there was generally a low level of awareness of *H. pylori*, with only 24% of the participants reporting they had ever heard of the infection, a value very similar to the percentage who had experienced an endoscopy. Similarly, only 13% of the participants remembered ever receiving an *H. pylori* test. Interestingly, there was a trend where individuals testing negative had heard of *H. pylori*, had been tested before this study, and had had follow-up treatment (e.g., endoscopy). The team previously reviewed the literature for studies on knowledge about *H. pylori* among patients and the general population, finding only nine papers [43]. Among them, one study also found higher knowledge of *H. pylori* among those testing negative and that a higher proportion of those testing negative had been tested before [44]. While it remains to be investigated, it seems possible that *H. pylori* knowledge is higher among those who have been treated before. However, these low levels of community awareness and prior testing compared to the *H. pylori* prevalence suggest that there need to be public campaigns directed to both the community and to health care providers in order to reduce or eradicate the infection.

This study has several strengths. First, few studies examining *H. pylori* infection among Indigenous or AI/AN populations in the US have been performed. Results from this study contribute to knowledge of prevalence and potential risk factors of infection in Navajos. The community was receptive to participation in this study, although work on the Navajo Nation had inherent challenges. Over 3000 miles were driven to acquire approval from the three chapters, two agencies and the NNHRRB, in addition to the mileage required for the ground-truthing of households and the recruitment. Second, nearly all participants recruited for the study provided complete data on all assessments. Thus, additional analyses to account or adjust for missing data were not needed. Third, only those risk factors found identified to be most relevant, salient, and meaningful to the Navajo communities sampled for this study were examined in analyses. Lastly, the mixed effects logistic regression undertaken accounted for correlation between risk factors among individuals living within the same household.

This study also had limitations. First, sample size was small and community samples were limited to one region of the Navajo Nation, which could limit generalizability of results. Second, *H. pylori* infection and colonization can persist throughout an individual’s lifetime. It is likely that adults in this study acquired *H. pylori* in childhood or at an earlier time point. Therefore, risk factors identified in this study may not accurately reflect social, environmental, or clinical conditions that immediately preceded acquisition of *H. pylori* infection. The estimated prevalence of *H. pylori* in this study did not include individuals under the age of 18, which could overestimate the prevalence of *H. pylori* infection in this target population. Lastly, according to the 2010 Census, nearly 51% of the on-reservation population of the Navajo Nation was 0–29 in age [45]. This age range was not adequately captured in this study, as only 11 individuals recruited and consented for this study fell within the 18–29 age range. Recruitment of individuals in the 18–29 age range was difficult largely because these individuals were often unavailable due to work (often off-reservation) during the work week and often unavailable on weekends. Future studies should explore different methods to recruit and consent the young working population on the Navajo Nation, perhaps catching them at community gatherings or during lunch breaks. Finally, while all participants were notified of their *H. pylori* status along with recommendations of seeking care, we did not monitor who went in for consultation and/or treatment. Future studies are needed to address barriers and help facilitators diagnose and treat *H. pylori* infection.

## 5. Conclusions

*H. pylori* prevalence is high among Navajo adults living northern Arizona, with prevalence estimates similar to those found in Alaska Natives. Use of unregulated household water sources, which include water from wells, springs, and community spigots, is strongly associated with presence of active *H. pylori* infection. In contrast to the high prevalence, the level of awareness of and testing for *H. pylori* is low. These findings support continued investigation into *H. pylori* among the Navajo Nation population and they identify a pressing need to address socioeconomic conditions, water supplies, and community awareness of gastric cancer. Better elucidation of risk factors is needed to develop strategies for eradication of the infection and earlier diagnosis of gastric cancer.

## Figures and Tables

**Figure 1 ijerph-19-00797-f001:**
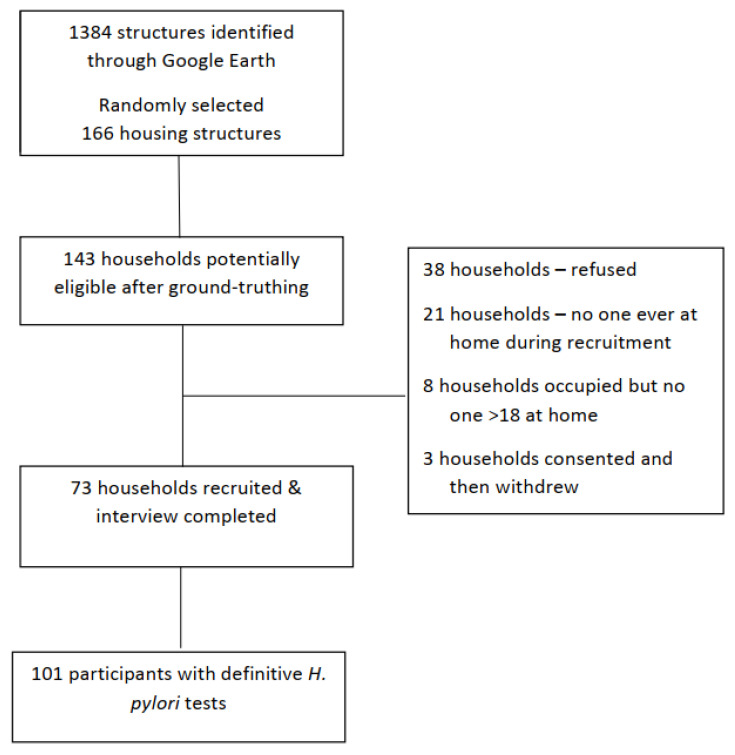
Flow of household recruitment to the Navajo Healthy Stomach Project.

**Table 1 ijerph-19-00797-t001:** Characteristics of participants in the Navajo Healthy Stomach Study by urea breath test (UBT) result.

	Overall *n* = 101	UBT Positive *n* = 66	UBT Negative *n* = 35	
Characteristics	*n* (%)	*n* (%)	*n* (%)	*p* ^‡^
**Sex**				0.03
Male	41 (40.6)	32 (48.5)	9 (25.7)	
Female	60 (59.4)	34 (51.5)	26 (74.3)	
**Age,** years				0.46
18–29	11 (10.9)	8 (12.1)	3 (8.6)	
30–49	26 (25.7)	19 (28.8)	7 (20.0)	
50–64	24 (23.8)	13 (19.7)	11 (31.4)	
65–79	30 (29.7)	18 (27.3)	12 (34.3)	
80+	10 (9.9)	9 (12.1)	2 (5.7)	
**Age,** years, dichotomized				0.22
<50	37 (36.6)	27 (40.9)	10 (28.6)	
≥50	64 (63.4)	39 (59.1)	25 (71.4)	
**Education**				0.78
<High School	35 (34.7)	22 (33.3)	13 (37.7)	
≥High School/GED	64 (63.4)	42 (63.6)	22 (62.9)	
Missing	2 (2.0)	2 (3.0)	-	
**Clinical History**				
Ulcers	4 (4.0)	2 (3.0)	2 (5.7)	0.51
Gallstones	25 (23.8)	14 (21.2)	10 (28.6)	0.41
Gastroscopy/Endoscopy (ever)	24 (24.0)	12 (18.18)	12 (34.29)	0.08
**Water Sources**				0.02
Regulated (piped, bottled)	75 (74.3)	43 (65.2)	32 (91.4)	
Unregulated (spring, well)	18 (17.8)	16 (24.2)	2 (5.7)	
Mixed	8 (7.9)	7 (10.6)	1 (2.9)	
**Have refrigerator in home**	94 (93.1)	60 (90.9)	34 (97.1)	0.33
**Travel > 1 h for groceries**	86 (85.1)	56 (84.8)	30 (85.7)	0.80

Note: *p*
^‡^ values were determined by Χ^2^ tests comparing UBT positive and negative individuals.

**Table 2 ijerph-19-00797-t002:** Association between risk factors and prevalence of *H. pylori* infection as determined by urea breath test (UBT).

Characteristics	UBT Positive Pos/Total (%)	Univariate	Model 1 ^‡^	Model 2 ^§^
		OR (95% CI)	AOR (95% CI)	AOR (95% CI)
**Overall**	66/101 (65.3)			
**Sex**				
Female	34/60 (56.7)	Ref	Ref	Ref
Male	32/41 (78.0)	2.72 (1.11–6.68)	3.10 (1.10–8.72)	3.26 (1.05–10.07)
**Age** years, dichotomized				
<50	27/37 (73.0)	Ref	Ref	Ref
≥50	39/64 (60.9)	0.58 (0.24–1.40)	0.37 (0.13–1.10)	0.36 (0.11–1.15)
**Education**				
<High School	22/35 (62.9)	Ref	Ref	Ref
≥High School/GED	42/64 (65.6)	1.13 (0.48–2.66)	1.20 (0.40–3.61)	1.20 (0.38–3.75)
**Clinical History**				
Ulcers	2/4 (50.0)	0.53 (0.07–3.83)	0.29 (0.03–3.03)	0.28 (0.02–3.23)
Gallstones	14/24 (58.3)	0.67 (0.26–1.73)	1.44 (0.47–4.38)	1.45 (0.46–4.61)
**Water Sources**				
Regulated	43/75 (57.3)	Ref	Ref	Ref
Unregulated	16/18 (88.9)	5.95 (1.28–27.76)	8.61 (1.45–51.05)	9.32 (1.35–64.51)
Mixed	7/8 (87.5)	5.21 (0.61–44.48)	5.97 (0.64–55.36)	6.38 (0.60–67.70)

Note: Abbreviations: UBT, Urea Breath Test; OR, Odds Ratio; CI, Confidence Interval. Model 1 ^‡^: Logistic regression adjusted for other variables in the table. Model 2 ^§^: Model 1 adjusted for other variables and with household as random effect.

**Table 3 ijerph-19-00797-t003:** Differences in self-reported *H. pylori* awareness by *H. pylori* infection status as measured by urea breath test (UBT).

	Overall *n* = 101	UBT Positive *n* = 66	UBT Negative *n* = 35	
	*n* (%)	*n* (%)	*n* (%)	*p*
Ever heard of *H. pylori*	24 (23.8)	12 (18.18)	12 (34.29)	0.07
Any family told they have *H. pylori*	11 (10.9)	6 (9.09)	5 (14.29)	0.43
Ever been tested for *H. pylori*	13 (12.9)	3 (4.54)	10 (28.57)	<0.01
Ever had a gastroscopy/endoscopy	24 (24.0)	12 (18.18)	12 (34.29)	0.08

*p* values were determined by Χ^2^ tests.

## Data Availability

The data are not publicly available due to requirements of the Navajo Nation Human Research Review Board. Separate approvals would need to be obtained.

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
