# Peer review of "Helicobacter pylori Prevalence and Risk Factors in Three Rural Indigenous Communities of Northern Arizona"

_ijerph, 2022, doi:10.3390/ijerph19020797_

Round 1

Reviewer 1 Report

The original article entitled “Helicobacter pylori prevalence and risk factors in three rural Indigenous communities of Northern Arizona” is correctly written and shows in a scientifically significant way the relationship between the consumption of contaminated water and the risk of developing an infection caused by H. pylori. I have a few minor comments about the Introduction (see below). Despite their insignificance in the context of the entire manuscript, in the future I recommend that particular attention should be paid to this part of the article, because it is usually the one that determines the first reception.

Suggested corrections:

  • “the most common stomach infections” -> the most common bacteria producing stomach infections [or similar in meaning] (line 18)
  • “the most common bacterial infection” -> the most common bacteria producing stomach infections [or similar in meaning] (line 34)
  • “infection [10” -> infection [10] (line 41)
  • “several severe astrointestinal outcomes” -> several severe gastrointestinal outcomes (line 41)
  • “stomach or gastric cancer” -> please choose one of this expressions (line 42)
  • “75%[29] compared with” -> 75% [29] compared with (line 82)
  • “gastric cancer in t high-risk populations” -> gastric cancer in the high-risk populations (line 96)
  • Additionally, in a few places in the Introduction empty spaces can be seen. Please correct.

Future recommendations:

I believe that it would be interesting from an epidemiological point of view to directly demonstrate the presence of H. pylori in the tested water sources, e.g. with the use of genetic and fluorescence methods. Please consider this issue.

Author Response

Response to Reviewer 1: Thank you for your careful review of this manuscript. We have made the corrections that you noted and checked the document for other errors. We agree that further work on water supplies is appropriate and they are being evaluated in our subsequent studies.

Reviewer 2 Report

 The authors of the article with title "Helicobacter pylori prevalence and risk factors in three rural Indigenous communities of Northern Arizona", they  presented and developed in an appropriate way and its goal was achieved effectively. A little advice can be valid suggestion that could be added a brief mention on the pathogenic H. Pylori mechanisms action and on geographic areas of his antibiotic resistance in the United States. Also in online 41 fix square bracket reference [10] and the missing "g" (gastrointestinal).

Author Response

Response to Reviewer 2: Thank you for your careful review of this manuscript. We have made the corrections that you noted and checked the document for other errors.

We agree that more is needed on antibiotic resistance since little is known about resistance in these populations for the current therapies. This topic is a focus of ongoing work for us within this community, but the results are incomplete.

Reviewer 3 Report

It is an interesting study. The presentation is ok but needs some corrections to improve its readability. The methodology and the results sections look organized, but please consider the below points at the revision stage:

  1. The introduction has too many numeric values and detailed information from previous studies. Please generalize the information and remove too much detailed information from your introduction. Also, please include the objective and significance of your study in the introduction. It should be improved.
  2. The article has some typos and space errors. 
  3. The Tables are not organized properly. It needs a better organization. 
  4. The conclusion should be improved as well. 
  5. The reference styles are not correct. It must be according to the journal guidelines.

Author Response

Response to Reviewer 3: Thank you for your careful review of this manuscript. We have made the corrections that you noted and checked the document for other errors. We also made revisions to the introduction to reduce the details. The tables have all been modified for better organization and readability.